Review

 

Subject Area:
cellular biology/genetics/neuroscience/molecular biology

Keywords:
clonal expansion, mtDNA, ageing, disease, mutation, heteroplasmy

Authors for correspondence:
Doug M. Turnbull
e-mail: doug.turnbull@ncl.ac.uk
Amy E. Vincent
e-mail: amy.vincent@ncl.ac.uk

†These authors contributed equally.

# The rise and rise of mitochondrial DNA mutations

Conor Lawless, Laura Greaves†, Amy K. Reeve†, Doug M. Turnbull and Amy E. Vincent

Wellcome Centre for Mitochondrial Research, Clinical and Translational Research Institute, Faculty of Medical Sciences, Newcastle University, Newcastle NE2 4HH, UK

CL, 0000-0002-4186-8506; LG, 0000-0002-8071-5916; AKR, 0000-0002-4474-2204; DMT, 0000-0002-8878-9901; AEV, 0000-0002-0360-6644

How mitochondrial DNA mutations clonally expand in an individual cell is a question that has perplexed mitochondrial biologists for decades. A growing body of literature indicates that mitochondrial DNA mutations play a major role in ageing, metabolic diseases, neurodegenerative diseases, neuromuscular disorders and cancers. Importantly, this process of clonal expansion occurs for both inherited and somatic mitochondrial DNA mutations. To complicate matters further there are fundamental differences between mitochondrial DNA point mutations and deletions, and between mitotic and post-mitotic cells, that impact this pathogenic process. These differences, along with the challenges of investigating a longitudinal process occurring over decades in humans, have so far hindered progress towards understanding clonal expansion. Here we summarize our current understanding of the clonal expansion of mitochondrial DNA mutations in different tissues and highlight key unanswered questions. We then discuss the various existing biological models, along with their advantages and disadvantages. Finally, we explore what has been achieved with mathematical modelling so far and suggest future work to advance this important area of research.

## 1. Evolutionary aspects of mitochondrial genetics

The endosymbiotic origins of mitochondria in eukaryotic cells are apparent in the similarities between mitochondria and bacteria, such as the plasmid-like multi-copy circular genome that resides in the mitochondrial matrix. Mitochondria are thought to be descended from α-proteobacterium due to sequence similarities between the genome and mitochondrial DNA (mtDNA) [1]. In present-day humans the mitochondrial genome is approximately 16.5 kb and encodes only 37 genes. This genome has been greatly reduced over evolutionary time via transfer of genes to the nuclear genome [2] so that it now only encodes 2 rRNAs, 13 OXPHOS subunits and 22 tRNAs [3].

Unlike diploid nuclear DNA, mtDNA exists in a highly polyploid state within each cell. This means that mutations in the mtDNA can exist in a subset of the total cellular mtDNA, a state termed 'heteroplasmy'. mtDNA molecules are replicated during mitosis, in much the same way that nuclear DNA molecules are (defined as strict replication [4]), but mtDNA molecules are also continuously replicated independently of the cell cycle (defined as relaxed replication [4]), in a similar fashion to the replication of bacterial genomes. Molecules of mtDNA have been shown to be continuously selected at random for relaxed replication in pulse-chase thymidine analogue experiments in dividing mouse cells [5].

Packaged as nucleoids, mtDNA resides close to the inner mitochondrial membrane and the oxidative phosphorylation (OXPHOS) complexes. The mitochondrial genome is hyper-mutable compared with nuclear DNA and this is thought to be due to damage caused by the high levels of reactive oxygen

royalsocietypublishing.org/journal/rsob    Open Biol. **10**: 200061

species (ROS) to which it is exposed [6,7], as well as the fact that mtDNA is replicated more frequently. Further, the compact nature of mtDNA means that there is very little that is non-coding and therefore mutations arising in mtDNA are much more likely to have pathological impact than mutations arising in the nuclear genome, where a large amount of the DNA is intronic.

Mutations in mtDNA can either be inherited or sporadically acquired throughout life. Inherited mutations are randomly segregated between primary oocytes and between different tissues. Work assessing mutation load in different fetal tissues has demonstrated that mutation load is similar across tissues [8]. Single cell analysis at this stage has not yet been completed, so we have no way of knowing whether the segregation of mutations during tissue formation contributes to the variability of mutation load or whether this is purely due to clonal expansion after development. In individuals who inherit a low mutation load, mitochondrial disease pathology is not likely to arise for a long time, if at all. The timing of the onset of mitochondrial disease is variable and pathology can be heterogeneous, affecting some tissues or parts of tissues or cells, but not others [9,10].

In comparison, sporadically acquired mutations arise in a single mtDNA molecule in a single cell and occur during healthy ageing, mtDNA maintenance disorders and a range of other diseases [11–13]. This single mutated mtDNA molecule is then either lost from the cell or clonally expands to higher levels. There is a tissue-specific pattern to the clonal expansion of sporadic mtDNA mutations: the accumulation of mtDNA point mutations is more common in mitotic cells whereas the accumulation of mtDNA deletions is more common in post-mitotic cells. Up to three or four different mtDNA deletions have been found to have clonally expanded in single muscle fibres and neurons respectively [13–15], although 37 mtDNA deletion species have been detected in a single neuron by ultra-deep sequencing [13].

## 2. Clonal expansion of mtDNA mutations

The dynamic process by which mtDNA mutations accumulate, which we often term 'clonal expansion', is thought to be one of the contributing factors behind the progress of many forms of mitochondrial disease. Furthermore, evidence suggests that the functional consequences of having clonally expanded mtDNA mutations may contribute to pathogenicity in other age-related diseases such as Parkinson's disease [16]. As such, this has become a very important question for mitochondrial biologists to answer, since it presents a potential therapeutic avenue that can be applied across a range of diseases.

### 2.1. Clonal expansion theories

One of the biggest unanswered questions regarding clonal expansion is whether there is any selective advantage or driver for the accumulation of mtDNA mutations. There have been a series of theories developed to explain how clonal expansion of mtDNA mutations occurs, with the random genetic drift theory assuming no selective pressures [17,18], and 'survival of the smallest' [19], 'survival of the sickest' [2,20], the negative feedback loop [21,22] and the 'perinuclear niche' [23] all assuming different possible selective mechanisms (figure 1). *In silico* predictions have been made from several of these theories

using estimations of important biological variables such as mtDNA copy number and mitochondrial turnover. All models have their limits but provide a means to test the degree to which a hypothesis can explain the levels of mitochondrial dysfunction that are observed in human samples.

So far evidence suggests that random genetic drift by relaxed replication is sufficient to explain the clonal expansion of point mutations but not necessarily mtDNA deletions. Random genetic drift suggests that mtDNA molecules are selected randomly for strict or relaxed replication leading to an accumulation of mutated mtDNA by chance. Stochastic, dynamic simulation models of mtDNA population dynamics during relaxed replication were previously developed [17,18,24] to explain clonal expansion as a form of random genetic drift [25]. Random genetic drift, which could explain clonal expansion without relying on any selective advantage or feedback at all, has formed a useful null hypothesis for testing theories about clonal expansion for the past 20 years.

In comparison with mtDNA point mutations, the clonal expansion of mtDNA deletions does not seem to be fully explained by random genetic drift, since results from random genetic drift modelling work to date do not accurately predict the levels of mtDNA deletions observed in substantia nigra neurons and muscle fibres (discussed further below). As such alternative hypotheses that include selective pressures have been considered. The first theory suggested to explain how mtDNA deletions expand clonally, suggesting that deleted, and therefore smaller, mitochondrial genomes would be replicated more quickly and so would have a selective advantage over wild-type genomes [19]. An alternative hypothesis suggests that the driving mechanism is a link between mtDNA encoded protein products and mtDNA replication [21,22]. *MT-ND4*, *MT-ND5* and *MT-ND6* were proposed as possible candidates for such a feedback mechanism with an mtDNA deletion that encompassed these genes leading to a decrease in their proteins and a compensatory increase in mtDNA transcription and replication to replenish these proteins. This increased replication would therefore provide a possible selective advantage for the deleted mtDNA. Subsequently observations in muscle have suggested that close proximity to nuclei provides a similar feedback mechanism for mtDNA deletions upregulating mtDNA replication, but through retrograde stress signalling for example changes in ATP/ADP or NAD$^+$/NADH ratios [23]. Both the positive transcriptional feedback loop and perinuclear niche hypothesis could feasibly contribute to clonal expansion of mtDNA point mutations also, however evidence suggests these selective pressures are not needed for clonal expansion. Similarly, it also remains possible that predictions from an adjusted random genetic drift model, fully incorporating uncertainty about parameters such as mtDNA copy number, mtDNA turnover and mitochondrial dynamics, will not be significantly different from observations in human samples.

## 3. Population dynamics of mtDNA point mutations

### 3.1. Formation of point mutations

The mechanism by which mtDNA point mutations are generated in somatic tissues is an area of much debate. Initial

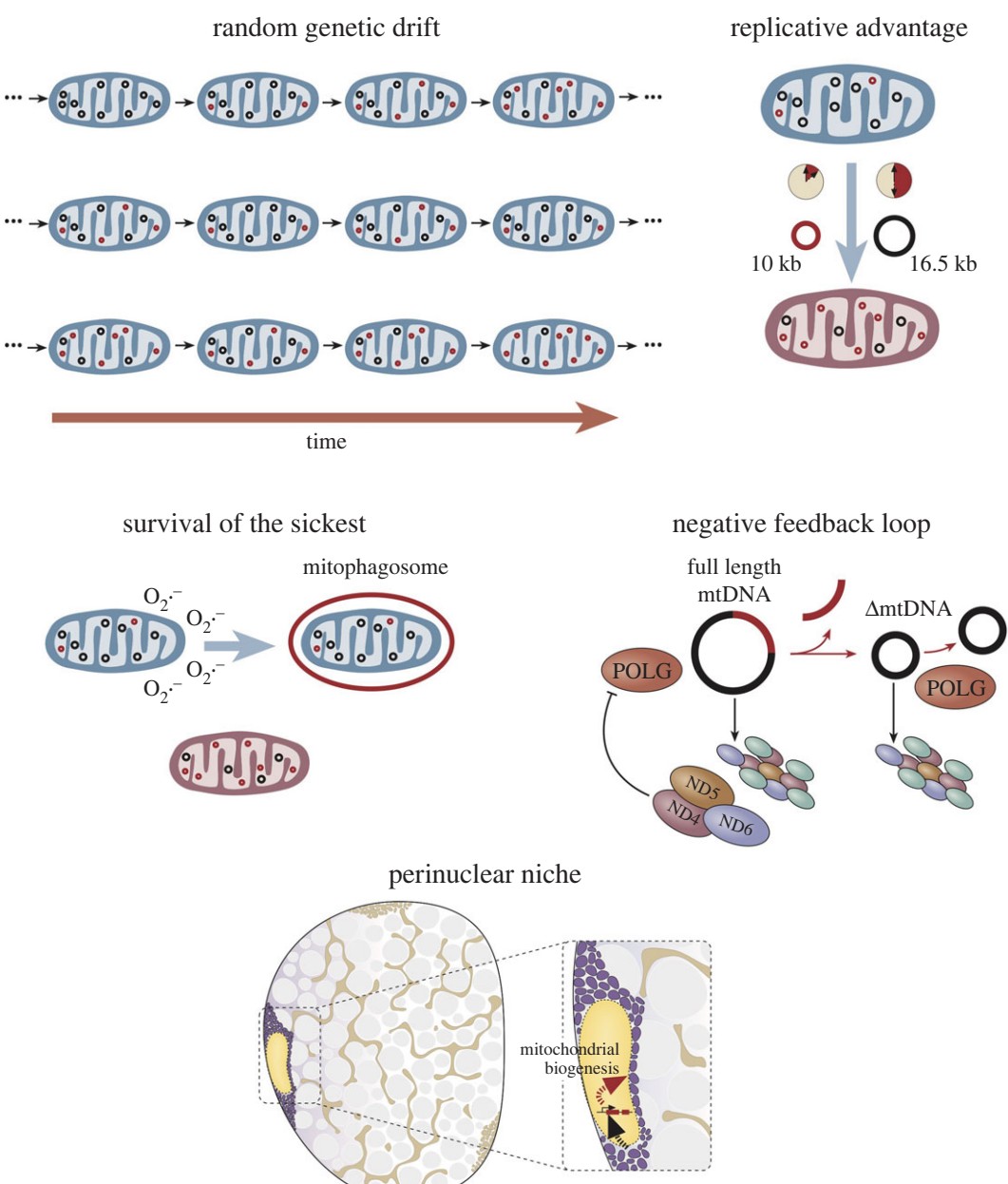

**Figure 1.** Theories of clonal expansion. The mechanism by which clonal expansion happens has been proposed to be explain by a number of possible theories about the mechanism. Random genetic drift suggests the accumulation happens randomly during relaxed mtDNA replication. A replicative advantage suggests that smaller deleted mtDNA genomes are replicated faster and take over the cell. Survival of the sickest suggests that the most dysfunctional mitochondria survive mitophagy due to a reduced production of ROS and therefore accumulate. A negative feedback look suggests a reduction in a mtDNA encoded protein product from a mutated mtDNA molecule drives further transcription and replication. Finally, the perinuclear niche hypothesis suggests a localized upregulation of mitochondrial biogenesis is triggered by mtDNA mutations accumulating adjacent to the myonuclei in muscle fibres.

hypotheses suggested that mtDNA was susceptible to damage induced by ROS due to its close proximity to the respiratory chain in the mitochondrial matrix [26]. ROS induced DNA damage causes base modifications, double and single-strand breaks, sugar damage and abasic sites [27]. The most commonly reported base lesions are thymine glycols and 7,8-dihydro-8-oxo-2′-deoxyguanosine [8-oxo-dG] [28]. 8-oxo-dG is thought to be the most mutagenic lesion, which can cause the mtDNA polymerase to mis-incorporate an A base opposite the oxidized G resulting in a G:C to T:A transversion following a second round of replication [28]. More recent analysis of mtDNA mutational spectra in ageing cells has revealed that it is not consistent with the predicted ROS induced damage pattern, instead the most commonly reported mutations are G:C to A:T transitions [29,30]. Transitions are more likely to

be the result of errors of the Polγ, which is responsible for replication of the mtDNA, and/or spontaneous cytosine deamination to uracil which then mis-pairs with adenine resulting in the G>A transition [31]. These data, alongside analysis of mtDNA point mutation spectra in transgenic animals [32], suggest there is a limited role for ROS-induced mtDNA mutagenesis causing point mutations in somatic cells.

## 3.2. Sporadic mtDNA point mutations

Sporadically acquired mtDNA point mutations, are the most common clonally expanded mutations in mitotic tissues. These were first detected by Nekhaeva *et al.* [33], who sequenced individual cells from normal ageing buccal epithelium and found that a subset of cells contained mtDNA point mutations

at high levels. Following this, Taylor *et al.* showed that a significant proportion of ageing colonic epithelial crypts (on average 15% by the age of 70) demonstrated histochemical loss of cytochrome *c* oxidase activity, which was caused by somatic, clonally expanded mtDNA mutations [10]. This work was followed up by a number of studies demonstrating an age-related increase in the frequency of cells with OXPHOS defects due to clonally expanded mtDNA mutations in mitotic tissues including the small intestine, stomach, oesophagus, prostate and liver [34–37].

Computational modelling of clonal expansion within mitotic cells has shown that random genetic drift is sufficient to explain the population dynamics of mtDNA point mutations observed experimentally in individual cells over time [24,38]. Random genetic drift models suggest that clonal expansion of a mutated mtDNA molecule to high levels within a cell is a relatively slow process and that initial mutational events must occur early in life. These early life mutations then either propagate randomly through mtDNA replication and segregation at cell division, or are lost. Successive cycles of this stochastic process allow mutated mtDNA molecules to become the dominant species within some cells, resulting in a mosaic pattern of cellular OXPHOS defects which are seen in ageing human mitotic tissues. In addition, most mitotic tissues have a high turnover rate and are maintained by long-lived stem cells. mtDNA mutations which expand clonally within these stem cells will then be propagated in their progeny. In tissues such as the liver, prostate, stomach and colon, this results in large, clonal patches of cells with identical mtDNA mutations and associated OXPHOS defects [34,36,38,39].

Despite numerous studies providing evidence that mtDNA mutations expand clonally causing a mosaic pattern of OXPHOS defects in ageing human tissues (e.g. colon; figure 2), the functional consequences of these have not yet been fully elucidated. A study by Nooteboom *et al.* showed that colonic crypts with loss of MTCO1 protein expression had fewer actively proliferating cells and were significantly smaller than those with normal MTCO1 levels, however whether this further effects tissue function is unknown [40]. Further insights have been possible through the development of mouse models. The mtDNA mutator mouse has a D257A amino acid change in the proof-reading domain of Polγ resulting in an error prone polymerase, causing an accelerated acquisition of mtDNA point mutations [41,42]. These mice show a premature ageing phenotype and significantly reduced lifespan. A mosaic pattern of OXPHOS defects due to clonally expanded mtDNA mutations have been shown in the small intestine [43] and the colon of these mice [44]. In the small intestine these defects have been shown to cause an increase in the frequency of apoptosis, a decrease in cell proliferation, a reduction in dietary fat absorption and a loss of the capacity to form stem cell derived organoids *in vitro* [45]. Analysis of the haematopoietic compartment of the mutator mice revealed no direct effect of mitochondrial dysfunction on the haematopoietic stem cells themselves, but instead defects during early differentiation were noted [46]. These differentiation blocks resulted in abnormal myeloid lineages and caused anaemia and lymphopaenia in the mice [47].

### 3.3. Inherited mtDNA point mutations

In contrast to the accumulation of somatic mtDNA mutations in ageing tissues, some inherited mtDNA point mutations are found to be systematically lost throughout life in rapidly dividing cells including blood, buccal mucosa and colonic epithelium, while remaining stable in post-mitotic tissues [48–52]. However, it is not known whether the decline in mtDNA mutation load in rapidly dividing cells is also the result of random genetic drift or the result of active selection against functionally deleterious variants (e.g. due to the loss of entire affected cells).

# 4. Population dynamics of mtDNA deletions

## 4.1. Deletion formation

mtDNA deletions have been proposed to form either during replication or repair. Features of mtDNA breakpoints have previously been used to infer the mechanism of formation, with deletions defined as class I if they have direct repeats, class II with indirect repeats and class III if they have no repeats [53]. Initially deletions were proposed to form during replication due to a slip-replication mechanism [54]. This model assumes that mtDNA is replicated by the strand-asynchronous mechanism and that the light strand misaligns so that the 3′ repeat of the light strand anneals to the 5′ end of the heavy strand, generating a single strand loop, this loop would be susceptible to a single strand break and degradation [54,55]. An alternative hypothesis was later proposed, which suggests mtDNA deletions are formed during mtDNA repair of double strand breaks, by the annealing of homologous repeats created by exonuclease activity at double strand breaks [53,55].

To further investigate replication-dependent mechanisms for deletion formation, a mouse model with an inducible mitochondrially targeted restriction endonuclease [56] led to the suggestion that mtDNA deletions are formed during mtDNA repair of double-strand breaks. In these mice double-strand breaks were induced in adult neurons, resulting in the formation of deleted mtDNA molecules. Evidence suggests that double-strand breaks may be repaired either by non-homologous end joining or micro-homology-mediated end joining [55–58]. Both of these mechanisms only account for homology-dependent recombination, whereas there are reports of deletions without repeat sequences [17,53]. More recent work looking at replication-dependent mechanisms has suggested that mtDNA deletions are instead formed by copy-choice recombination during active L-strand DNA synthesis [59]. This model is attractive since mtDNA deletions with direct repeats, imperfect repeats and no repeats have been detected and while this mechanism is enhanced by the presence of mtDNA repeats it can also occur without repeats. As such it is now thought that mtDNA deletions form either by copy choice recombination during mtDNA replication or by repair of double strand breaks via non-homologous end joining or micro-homology mediated end joining [60]. Furthermore, the mechanism for deletion formation may be dependent on the mechanism of mtDNA replication, which has been demonstrated to vary between tissues [61].

## 4.2. Sporadic mtDNA deletions

In contrast to mtDNA point mutations, sporadically acquired mtDNA deletions are the most common sporadic, clonally expanded mtDNA mutation in post-mitotic cells. The reason

royalsocietypublishing.org/journal/rsob    Open Biol. **10**: 200061

|  | COX positive | intermediate | COX deficient |
| --- | --- | --- | --- |

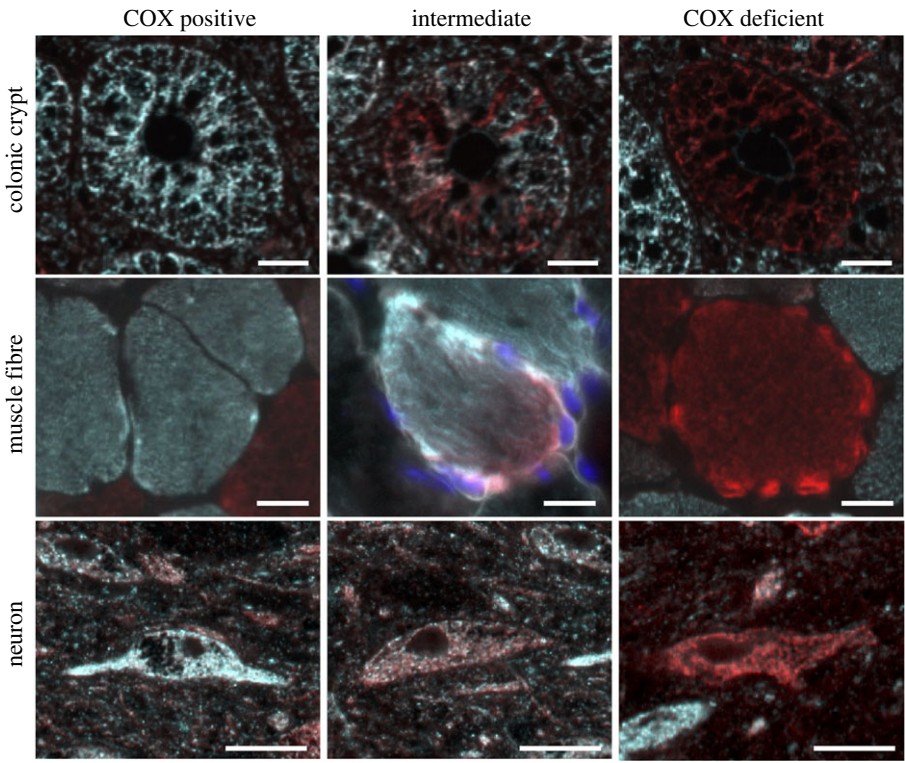

**Figure 2.** Consequences of clonally expanded mitochondrial DNA mutations. Mitochondrial DNA mutations accumulate with age and disease in both mitotic tissues such as the colon, and post-mitotic cells such as muscle fibres and neurons. At low levels the mitochondrial DNA mutation will have little functional impact (COX positive). As the percentage of mitochondrial DNA mutations within a cell accumulate they will exceed a biochemical threshold causing mitochondrial respiratory chain dysfunction (COX deficient). The transition from COX positive to COX deficient is slightly different in different tissues, with the most noticeable differences being that colonic crypts can be partially COX deficient, muscle fibres can have focal deficiency, whereas neurons are more commonly observed to have low deficiency across the full cell body (although it is never possible to view a full neuron). Scale bar, 25 μm. The images were collected from the authors' own research; see ethics statement at end of article for further details.

for this contrast remains an important unsolved mystery. High levels of clonally expanded mtDNA deletions have been reported in neuronal populations from the substantia nigra, hippocampus, striatum and spinal cord with both ageing and disease [12,62–64], as well as aged skeletal muscle fibres [65]. These mtDNA deletions accumulate to high levels in both neurons and muscle fibres, leading to mitochondrial dysfunction. However, the mutation loads observed in different neuronal populations differ substantially, with the highest levels (over 50%) typically observed in the dopaminergic neurons of the substantia nigra [66,67].

Extensive studies have demonstrated that clonally expanded mtDNA deletions are associated with mitochondrial OXPHOS dysfunction in both neurons and muscle fibres [9,65] (figure 2). Furthermore, in the brain these have been found to be associated with neurodegeneration [62], and there is evidence that they are associated with muscle fibre atrophy [65]. Despite increasing knowledge about the consequences of clonally expanded mtDNA deletions, it is not yet fully understood how these deleted mtDNA species come to predominate in cells.

While much is known about mitochondrial genetics, the process by which a single mtDNA mutation accumulates still eludes us. Based on data from estimates of mtDNA turnover rate and mtDNA copy number from rat muscle fibres, modelling of random genetic drift by Elson *et al.* [18] predicts 4% of post-mitotic cells should become COX-deficient over an 80 year period in healthy individuals. However, that prediction does not include a range of uncertainty or confidence interval.

Data from human substantia nigra shows that respiratory chain deficiency is higher with 40% of neurons found to be

COX-deficient at 80 years [68,69], and this is likely to be an underestimate since approximately 5% of neurons are lost per decade [70]. Levels of COX-deficient cells in a single section of aged muscle are much lower with percentage of COX-deficient fibres typically less than 5% in people aged 75–88 years [71]. However when the length of muscle fibres in the quadriceps vastus lateralis is considered, the true percentage may be higher, approximately 6% at 49 years and approximately 31% by the age of 92 years [65]. Again, there is considerable unstated uncertainty about these predictions.

However, as well as assuming random genetics, the model of Elson *et al.* [18] also assumes that fibres are well mixed (i.e. it ignores diffusion of mtDNA along the considerable length of muscle fibres). If this model of random genetic drift underestimates the percentage of respiratory chain deficient cells, it may be that there are selective pressures at work that are preferentially allowing the accumulation of these deleted mtDNA species.

An alternative to purely random drift is that deleted mtDNA molecules have a replicative advantage [19]. Many studies *in vitro* have generated supporting evidence for this theory, finding that deleted mtDNA molecules repopulate cells faster than wild-type mtDNA under relaxed copy number control [72,73]. However, following artificial depletion of the mtDNA, the cell may replicate its mtDNA at a faster rate in order to return to the required mtDNA copy number, under such conditions the time taken to replicate a single molecule could be much more rate limiting and therefore important, than it would be in post-mitotic cells. For example, work in human muscle has demonstrated no relationship between the size of mtDNA deletions and respiratory chain deficient

royalsocietypublishing.org/journal/rsob Open Biol. **10**: 200061

segments, suggesting that there is no replicative advantage [9]. However, we do not know when each of the mutations formed, but must assume that if deletions form randomly with a mix of small and large deletions this should not impact results. Finally, work in *C. elegans* examining the behaviour of two differently sized mtDNA deletions, also found no evidence for a replicative advantage [74].

Another alternative hypothesis that has been proposed is a feedback mechanism whereby low levels of protein products from the deleted genes, triggers mtDNA replication [21,22]. While this hypothesis is attractive, the authors suggest a subset of 'feedback genes'. However, examination of the spectrum of breakpoints reported on 'mitobreak' [75] demonstrates that no one gene is always deleted and indeed deletions occur that do not remove any of the suggested genes. Therefore, if such a mechanism exists, it is likely that the number of genes contributing to the feedback mechanism is either larger or all inclusive.

More recently, we have proposed a perinuclear niche hypothesis for the clonal expansion of deletions in skeletal muscle. This stems from the observation that the smallest regions of respiratory chain deficiency are both subsarcolemmal and perinuclear. The hypothesis suggests that the close proximity of the mtDNA deletion and dysfunctional mitochondria to the nucleus provides a driver of clonal expansion through retrograde stress signalling triggering a local increase in mtDNA replication. Such stress signalling may include a reduction in ATP/ADP or NAD+/NADH ratios, upregulation of the integrated mitochondrial stress response [76] or (as suggested in the original investigation) upregulation of mitochondrial biogenesis via the unfolded protein response [23]. Furthermore, if mtDNA replication is higher in the perinuclear region of muscle fibres as previously reported in HeLa cells [77], this may also lead to a higher frequency of replication errors and deletion formation in the perinuclear mitochondria. The perinuclear hypothesis would likely favour sporadic mutations arising in close proximity to the nucleus, and therefore it is possible that it would also favour mtDNA point mutations. However, point mutations are less commonly investigated in muscle and further work would need to be completed to investigate this.

This hypothesis only suggests a selective pressure after the mtDNA deletion has reached sufficient levels locally to cause respiratory chain deficiency, and it has yet to be determined whether such an advantage is sufficient to explain the high levels of respiratory chain deficient cells observed. However, it is also possible that such focal deficiency could form without an induction of replication if mtDNA replication is naturally higher in the perinuclear region, as previously suggested [77]. In muscle, the mitochondria are either packed around the edge of the fibre adjacent to the myonuclei or between the myofibrils [78] and as such transport is minimal with mitochondrial fission and fusion providing the main means for distribution of mtDNA and proteins throughout the cell [79–81]. These attributes of mitochondrial organization and dynamics, as well as muscle fibre structure are integral to the perinuclear hypothesis. Therefore while we cannot yet rule out the existence of a similar mechanism in neurons, it is possible that an alternate mechanism may be present in neurons, in spite of perinuclear replication, given the more dynamic nature of the mitochondria in this cell type.

In comparison with muscle, the mitochondria in neurons must be transported from the site of biogenesis to areas of high energy demand (e.g. the synapse and nodes of Ranvier) and subsequently to the site of degradation. The majority of mitochondria are found to be stationary at the sites of ATP requirement, with around 10–20% of mitochondria being actively transported [82]. This small proportion of moving mitochondria and the sites at which mitochondrial biogenesis and degradation occur are likely to be important factors when we are considering how mtDNA mutations clonally expand in neurons. Previously, it has been demonstrated that when mitochondria lose their membrane potential they are transported back to the cell body for degradation [83,84], this is further supported by a lack of mitophagy observed in dendritic arbours and axonal projections in mitoQC mice, with the majority of mitochondrial degradation observed in the soma [85]. Furthermore, if the majority of mtDNA replication occurs in close proximity to the nucleus as previously reported in cultured cells [77], it is likely that the site of clonal expansion is in the cell body with mtDNA mutations being distributed along the neuron by mitochondrial transport, fission and fusion.

In muscle the presence of perinuclear foci of mitochondrial dysfunction suggests that the nuclei play an important role in the accumulation of mtDNA deletions and mitochondrial dysfunction. Therefore, the perinuclear niche hypothesis shows promise for explaining how these mtDNA deletions accumulate. However, it will be necessary to understand the relative contributions of mtDNA replication and mito-nuclear signalling to this process and to systematically compare this to random genetic drift using *in silico* modelling (discussed in more detail below). The structure of neurons would favour a situation where the majority of mitochondrial replication would occur in the perinuclear area and thus would also support a similar perinuclear niche hypothesis for the clonal expansion of mtDNA deletions to that proposed for muscle. However, this is less definitively described in neurons than muscle and also needs to take into consideration the vastly different movement and dynamics of mitochondria within these two cell types. This may explain why perinuclear focal deficiency has not to date been reported in neurons; however, it is still possible that such focal deficiency occurs and is simply less frequent or more challenging to find.

## 4.3. Single, large-scale mtDNA deletions

Similar to mtDNA point mutations, there is a disparity in what happens for inherited and sporadically acquired mtDNA deletions. Inherited mtDNA deletions such as those in patients with Pearson's syndrome are also lost from the blood [86]. However, it does appear in post-mitotic tissues that inherited mtDNA deletions loads are maintained throughout life or clonally expand. High levels of mtDNA deletions have been detected in individual muscle fibres and are associated with respiratory chain deficiency [87]. High levels of inherited mtDNA deletions have also been detected in neurons [88].

## 5. Important challenges and unanswered questions

Despite interest in understanding clonal expansion during development and ageing, and in disease, there are several important challenges that have hindered progress in understanding this important mechanism. Furthermore, there are

royalsocietypublishing.org/journal/rsob    Open Biol. **10**: 200061

many unanswered questions which are inherently important for understanding clonal expansion.

## 5.1. How does clonal expansion occur during development?

As stated above we know that inherited mtDNA variants are segregated unevenly between oocytes and that in a foetus tissue homogenates have a similar mtDNA mutation load [8]. However we know little about the formation of somatic mutations *in utero*, or about the single-cell mutation loads in fetal tissues. The question of clonal expansion during development *in utero* presents many unknown parameters for which it is all but impossible for us to gather biological data independently, making it clear that there is an important role for mathematical modelling of our uncertainty about this process. We can build hypotheses about how inherited mutations expand from the initial condition of the progenitor oocyte and how new mutations arise during development and compare the predicted consequences (e.g. mutation load distributions in fetal samples) with observations. However, it is important to be explicit about our uncertainty about model parameters such as initial mutation load, replication rate and copy number control, many of which are difficult or impossible to measure directly, without assuming a model of how expansion happens.

## 5.2. Why do mitotic cells and post-mitotic cells differ?

As discussed above, somatic mtDNA point mutations tend to accumulate in mitotic cells, while somatic deletions predominate in post-mitotic cells. The key to why this difference in the appearance of mtDNA mutations exists is likely to lie in the differences between the two types of cell. In mitotic tissues, mtDNA undergoes both strict (frequent) and relaxed (infrequent) replication, whereas in post-mitotic tissues, only relaxed replication occurs. As such this means that mtDNA is more frequently replicated in mitotic cells than post-mitotic cells. In addition, in replicating cells mitochondria are segregated to different daughter cell populations resulting in an asymmetric or symmetric distribution of mutant mtDNA between daughter cells, allowing for a bottleneck effect impacting clonal expansion [38].

Cells in mitotic tissues are regularly turned over and undergo apoptotic cell death. In comparison, in multinucleated skeletal muscle fibres it is thought that apoptosis occurs for single nuclei within a muscle fibre leading to muscle fibre atrophy [89], although this hypothesis is contentious [90]. Satellite cells may also fuse with a muscle fibre allowing regeneration of the fibre [91]. Both of the regeneration and apoptotic processes are effectively sub-cellular, and as such, a whole cell (and indeed the whole population of mitochondria in a muscle fibre) is not removed on a periodic basis. In neurons where cell death occurs during neurodegeneration, there is limited means to replace cells that are lost. Some neuronal populations in particular show a steady decline overtime [70], it is not understood why these are lost but it may be these are the ones with the highest mutation load.

## 5.3. How do inherited and acquired mutations differ?

There are clear differences between the changes in sporadic mtDNA mutations and inherited mtDNA mutations over time. Inherited mtDNA mutations are lost in rapidly dividing

cells such as the blood, intestinal epithelium, buccal mucosa and urine [49–52,92]. In post-mitotic cells not much is known about what happens to inherited mtDNA point mutations overtime, but it is commonly believed that single, large-scale mtDNA deletions clonally expand from birth in skeletal muscle fibres [9] and neurons, with higher mtDNA deletion loads in post-mitotic tissues than in mitotic tissues of the same patient [93,94]. In comparison both somatic mtDNA point mutations and mtDNA deletions clonally expand over time. It has been hypothesized that, in neurons this accumulation of somatic mutations, may be due to a difference in how cells respond to a mutation that is acquired in comparison to one that has been inherited [95]. Such a hypothesis is intriguing and warrants investigation in other tissues.

## 5.4. How do we investigate a process that takes place over a lifetime?

One of the greatest problems is that the accumulation of mutated mtDNA to pathological levels usually occurs over decades in humans. However, the most direct observations we can make are cross-sectional observations in patient tissue. Furthermore, clonal expansion is dynamic and is heterogeneous, occurring independently in individual cells. In order to capture this heterogeneity, observation of many single cells is required. However, even if it were possible to revisit the same cell repeatedly *in vivo*, it is not possible to track all of the individual mtDNA molecules within a cell over time. Direct, longitudinal observation of clonal expansion is simply not possible.

One helpful approach has been to examine spatial patterns within tissues (e.g. subcellular location of respiratory chain deficiency). From such studies we can deduce presence and location of mutations in single cells [9,23], and make deductions or predictions as to how these arose. It is difficult to know what happened prior to the sample being collected and it is important that we incorporate that uncertainty into our predictions.

Given the practical constraints on making longitudinal observations throughout human lifespans, in order to understand clonal expansion, we need to work with a model system. Importantly, the goal of any model (mathematical, statistical or biological) is that it should be a simpler, more tractable version of the real system, capturing its main characteristics. Models do not need to replicate the target system exactly in order to be useful, provided the differences are considered when interpreting the results. It is important that we carefully choose models appropriate for the question at hand.

# 6. Modelling clonal expansion

## 6.1. Biological models

Examining mtDNA population dynamics in mitotic cell culture has a number of advantages, including the use of patient-specific human cells, which are an experimentally tractable model of clonal expansion during development. For example, with mitotic cell culture it is possible to observe point mutations arising, reaching loads as high as 25% and declining in continuously dividing cell culture in as little as 21 weeks [96]. However, there are substantial disadvantages to such culture based models, including: time scale (although, in principle, cells can be cultured almost indefinitely); the

possibility that culture conditions may alter the selection of mtDNA mutants versus wild-type; we are likely only to be able to observe expansion of point mutations in continuously dividing culture; and many aspects of cell biology (mitochondrial dynamics, mtDNA replication rate etc.) we observe in one cell type may not be representative of others *in vivo* in particular post-mitotic cells (e.g. skeletal muscle fibres, neurons). One alternative for this last point is the growing work using patient derived induced pluripotent stem cells (iPSCs) [97], which can be differentiated into any cell type and also used to generate 2D or 3D co-cultures that are more representative of tissues. Such cultures are technically difficult to establish and maintain, however, and still suffer from many of the disadvantages listed above.

Animal models have the advantage of providing physiologically relevant conditions and cell types. For practical reasons, mouse models are a popular choice, however many do not accumulate the same levels of respiratory chain deficiency as humans. Differences between mouse and human tissue probably arise because of their much shorter lifespan but could also be due to significant difference in cell types (e.g. fibre type composition in muscle) and possibly mtDNA population size. This is important to consider when interpreting findings from animal models since, for example, Elson et al. [18] demonstrated a slower increase of mtDNA mutation load when the cellular mtDNA population is larger. Another advantage of animal models is the ability to manipulate process or use techniques to label cellular components and thus test hypotheses. However, animal models still present similar issues to human tissue samples since sampling tissue from mice still requires cross-sectional observations at single time points.

In addition to mouse models, *Caenorhabditis elegans* has been used in several studies which looked to investigate the clonal expansion of deleted mtDNA molecules [74,98] and mtDNA point mutations [99]. *Caenorhabditis elegans*, similar to mice, has a shorter lifespan than humans and as such does not have sufficient time for sporadic mtDNA mutations to clonally expand in wild-type animals [100]. However it provides a cost-effective means to investigate some of the possible selective pressures that could impact clonal expansion from low mutation loads in physiologically relevant tissues, such as the activation of retrograde signalling responses, and changes in mitochondrial biogenesis or mitophagy [74,98].

## 6.2. Mathematical models

Work on mathematical models of clonal expansion has continued since Chinnery et al. first modelled relaxed replication demonstrating that heteroplasmy can shift quickly over a short period of time [17]. This work was developed further by Elson et al. to explain between-cell heterogeneity in mtDNA mutation load dynamics [18]. Kowald & Kirkwood add the process of transcription to the same underlying model [21]. Stamp et al. incorporate the effect of asymmetric division of mitotic stem cells along with random drift [38]. Johnston & Jones consider the effect of various copy number control schemes on theoretical mutation load distributions between cells [101].

As stated before, random genetic drift presents an important, simple null hypothesis to which, new models can be compared. Indeed, while it does seem to explain clonal expansion of mtDNA point mutations in colonic crypts, it does not seem to fully explain clonal expansion of mtDNA deletions in post-mitotic cells. However, it is possible and indeed

likely that this is due to the specific assumptions embedded in the modelling by Elson et al. [18] about fixed mtDNA copy number, copy number control, mtDNA mutation rate, mtDNA half-life and mixing of mtDNA populations within the cell, all of which we still know very little about. Furthermore, these parameters are likely to vary between different tissues and cell types, and so inferences based solely on information from other cell types and tissues should be avoided. In order to assess the predictions from the random genetic drift hypothesis, we need to take intrinsic stochasticity as well as uncertainty about parameter values and processes into consideration, ideally through formal statistical inference, using methods that can handle parameter uncertainty and stochastic models. Henderson et al. demonstrate the computational and analytical difficulties behind inference for a model of mtDNA population dynamics [102]. Indeed, Bayesian inference for stochastic simulation models remains an active area of research in applied statistics [103–105], but there are several tools currently available to help with exact Bayesian inference for deterministic models [106–108].

Building a mathematical model helps us to be explicit about our mechanistic hypothesis about how a system works. Simulating from that model helps us make predictions about its consequences. We can then test the hypothesis by comparing predictions with experimental observations.

We expect that in order to make progress, particularly on the difficult problem of understanding the expansion of mutations in post-mitotic tissue (e.g. muscle fibres and neurons), we will need to incorporate our considerable uncertainty about the process of clonal expansion into our analysis of simulation model output. To make robust, statistical comparisons between predictions and observations, we need to include uncertainty about parameters into simulations from a model, propagating that uncertainty forward to uncertainty about predicted outputs. However, direct experimental observations of parameters relevant to clonal expansion (particularly the rate of mtDNA replication) is not currently possible. Assuming a dynamic model and fitting its output to cross-sectional, single-cell observations of copy number distributions and mutation load distributions over time is an important approach to learn about these processes in human cells.

Models are useful for bridging the gap between what can be observed and the process of interest. In this case we can make cross-sectional observations of mtDNA mutation load and copy number in hundreds of heterogeneous single cells, sometimes at multiple time points. We can use mathematical models to simulate mtDNA replication, including selection processes and potentially cell loss, to predict what observed mutation load distributions would look like under different hypotheses. For example, Elson et al. [18] created a model based on the random drift hypothesis and demonstrated that, for a plausible set of parameters (mtDNA copy number, mtDNA replication rate, mtDNA mutation rate), over a human lifespan, the proportion of post-mitotic cells where mutations arose de novo and expanded clonally by neutral drift during relaxed replication increased significantly with age.

Writing a model to describe clonal expansion forces us to be explicit about the mtDNA population we have in mind. For example, the model simulations carried out by Elson et al. [18] are of random genetic drift during relaxed replication, in well-mixed cells, with a fixed rate of mutation, a fixed replication rate and a fixed copy number. In reality, all of these parameters are likely to change with tissue or cell type and might even

change in time and space. If we are to compare model predictions with spatial data (e.g. the spatial distribution of mutation loads along massive muscle fibres) then we need to update the model to include those spatial effects. Work by Elson *et al.* [18] and subsequent models (e.g. model of mtDNA population dynamics in mitotic epithelial crypts by Stamp *et al.* [38]) assume perfect control of mtDNA copy number. If copy number is not tightly controlled, then it is likely that mtDNA population dynamics will include nonlinear bottleneck effects.

The point here is that, even under the label 'random genetic drift hypothesis', there is quite a wide range of different ways to represent the mechanisms underlying this hypothesis. It is important to explore the full range of plausible mechanisms (formal model selection) as well as to explore the full range of plausible parameter values (formal parameter inference). In some cases, it is possible to frame the choice between different mechanisms or even different hypotheses as parameter inference. For example, including selective advantage into a model of random genetic drift, it could be possible to infer statistically that the parameter representing selective advantage is not significantly different from zero, thereby rejecting that hypothesis.

Experimental data derived from patient tissue are difficult to gather and are very valuable. By carrying out parameter inference for a specific model and validating its predictions by comparing with data [102], we make the best use we can of patient tissue; for example, we can use this approach to estimate tissue-specific mutation rates, mtDNA copy number, replication rates and the strength of any selective advantage, together with our uncertainty about them, even for individual patients. Although it is difficult work, requiring experimental design specifically targeted at a particular model and challenging computation, the payback is that we can infer estimates for parameters that are difficult to access as well as make and assess quantitative predictions of mutation load distributions (and their uncertainty) which we can use to assess the validity of models and their underlying hypotheses. Learning about long-term dynamic processes driving clonal expansion throughout human lifespans is extremely difficult. However, the availability of rich, single-celled datasets makes this approach more realistic and promising than ever before.

## 7. Conclusion and future perspectives

The ongoing revolution in technology allows single-cell observations of mtDNA populations and allows us to discriminate more precisely between hypotheses about clonal expansion. Single-cell analysis provides us with the opportunity to quantify the distribution of cellular outcomes within an individual patient. To further our understanding of clonal expansion, we should take advantage of these rich data by comparing directly with distributed model predictions. We should look to re-visit older hypotheses such as random drift, as well as newer ideas such as perinuclear niche hypothesis by building and assessing mathematical models. We would like to alert our computational biology colleagues that the impossibility of direct observation of clonal expansion makes this a field where modelling work will make a real and important contribution. While learning about clonal expansion, particularly in post-mitotic cells, the modelling, statistical and computational challenges will be difficult but rewarding. By understanding this puzzling phenomenon that holds important pathological relevance to a range of diseases, we can then start to look for ways to slow or indeed halt this process. Such a therapeutic target would have extensive applications; however, it will also be important to understand key differences between diseases, tissues and mutations if we are to hold any hope of developing treatments.

Ethics. Muscle biopsies from quadriceps were obtained via needle biopsy under local anaesthesia. Ethical approval was granted by the Newcastle and North Tyneside local research ethics committees (reference 2002/ 205), and prior informed consent was obtained from each participant. Formalin-fixed paraffin-embedded (FFPE) human midbrain tissue sections were obtained from the Newcastle Brain Tissue Resource (NBTR, https://nbtr.ncl.ac.uk/), with approval from the Local Research Ethics Committee and adherence to the Medical Research Council's (MRC) Guidelines on the use of human tissue in medical research. Colon tissue was collected during surgery with prior informed consent and was approved by the Joint Ethics Committee of Newcastle and North Tyneside Health Authority (2001/188) and the National Research Ethics Committee London-Stanmore (11/LO/1613).

Data accessibility. This article does not contain any additional data.

Authors' contributions. A.E.V., C.L., L.G. and A.K.R. contributed to drafting the manuscript. A.E.V. drafted the figures. All authors critically revised the manuscript and approved the final version.

Competing interests. We declare we have no competing interests.

Funding. This work was supported by the Wellcome Centre for Mitochondrial Research (203105/Z/16/Z). A.E.V. is in receipt of a Sir Henry Wellcome Fellowship (215888/Z/19/Z). A.K.R. is in receipt of a Senior Parkinson's UK Fellowship (F-1401).

Acknowledgements. The authors would like to thank Anna Smith and Dr Chun Chen for contributing the images of colonic crypts and neurons, respectively. We would also like to thank Paula Rutter for redrawing figure 1.

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
