## [Reviewer comments · Open Biology]

Review History

RSOB-20-0061.R0 (Original submission)

Review form: Reviewer 1

Recommendation

Accept with minor revision (please list in comments)

Do you have any ethical concerns with this paper?

No

Comments to the Author

The review provides a good overview of the current understanding of mtDNA accumulation in cells. It does a good job of providing information to the reader, but the writing is somewhat dry.

Our understanding of clonal expansion is still incomplete. This review summarizes the current models. It would be helpful if the authors pointed out if they favor a particular model, or otherwise help the reader to think about this problem.

Other points:

1) In first paragraph, the circular and multi-copy nature of mtDNA is pointed out as evidence of the endosymbiotic origin of mitochondria. It would be helpful to point out the sequence analysis

indicating phylogenetic similarity.

2) Page 6; It is pointed out that clonal expansion of mtDNA deletions cannot be explained by random drift. This idea is important and should be explained in more detail. What feature cannot be explained by random drift?

3) Page 6: in discussing the negative feedback model for mtDNA deletions, why can't this model be applied to point mutations that reduce expression of the same proteins?

4) Page 6: In discussing the nuclear proximity model--how would that mechanism favor mtDNA deletions? Why would genomes with deletions be near the nucleus?

5) Instead of simply stating the current models of clonal expansion, it would be helpful if the authors pointed out their favorite view and why. Otherwise the review seems rather dry and does not help the reader think about the problem.

Decision letter (RSOB-20-0061.R0)

16-Apr-2020

Dear Dr Vincent,

We are pleased to inform you that your manuscript RSOB-20-0061 entitled "The rise and rise of mitochondrial DNA mutations" has been accepted by the Editor for publication in Open Biology. The reviewer has recommended publication, but also suggest some minor revisions to your manuscript. Therefore, we invite you to respond to the comments and revise your manuscript.

Please submit the revised version of your manuscript within 7 days. If you do not think you will be able to meet this date please let us know immediately and we can extend this deadline for you.

Please see our detailed instructions for revision requirements
<https://royalsocietypublishing.org/rsob/for-authors>

1) A text file of the manuscript (doc, txt, rtf or tex), including the references, tables (including captions) and figure captions. Please remove any tracked changes from the text before submission. PDF files are not an accepted format for the "Main Document".

2) A separate electronic file of each figure (tiff, EPS or print-quality PDF preferred). The format

should be produced directly from original creation package, or original software format. Please note that PowerPoint files are not accepted.

3) Electronic supplementary material: this should be contained in a separate file from the main text and meet our ESM criteria (see <http://royalsocietypublishing.org/instructions-authors#question5>). All supplementary materials accompanying an accepted article will be treated as in their final form. They will be published alongside the paper on the journal website and posted on the online figshare repository. Files on figshare will be made available approximately one week before the accompanying article so that the supplementary material can be attributed a unique DOI.

Online supplementary material will also carry the title and description provided during submission, so please ensure these are accurate and informative. Note that the Royal Society will not edit or typeset supplementary material and it will be hosted as provided. Please ensure that the supplementary material includes the paper details (authors, title, journal name, article DOI). Your article DOI will be 10.1098/rsob.2016[last 4 digits of e.g. 10.1098/rsob.20160049].

4) A media summary: a short non-technical summary (up to 100 words) of the key findings/importance of your manuscript. Please try to write in simple English, avoid jargon, explain the importance of the topic, outline the main implications and describe why this topic is newsworthy.

Images

Data-Sharing

It is a condition of publication that data supporting your paper are made available. Data should be made available either in the electronic supplementary material or through an appropriate repository. Details of how to access data should be included in your paper. Please see <http://royalsocietypublishing.org/site/authors/policy.xhtml#question6> for more details.

Data accessibility section

Sincerely,

The Open Biology Team

<mailto:openbiology@royalsociety.org>

Reviewer's Comments to Author:

Referee:

Comments to the Author(s)

The review provides a good overview of the current understanding of mtDNA accumulation in cells. It does a good job of providing information to the reader, but the writing is somewhat dry.

Our understanding of clonal expansion is still incomplete. This review summarizes the current models. It would be helpful if the authors pointed out if they favor a particular model, or otherwise help the reader to think about this problem.

Other points:

- 1) In first paragraph, the circular and multi-copy nature of mtDNA is pointed out as evidence of the endosymbiotic origin of mitochondria. It would be helpful to point out the sequence analysis indicating phylogenetic similarity.
- 2) Page 6; It is pointed out that clonal expansion of mtDNA deletions cannot be explained by random drift. This idea is important and should be explained in more detail. What feature cannot be explained by random drift?
- 3) Page 6: in discussing the negative feedback model for mtDNA deletions, why can't this model be applied to point mutations that reduce expression of the same proteins?
- 4) Page 6: In discussing the nuclear proximity model--how would that mechanism favor mtDNA deletions? Why would genomes with deletions be near the nucleus?
- 5) Instead of simply stating the current models of clonal expansion, it would be helpful if the authors pointed out their favorite view and why. Otherwise the review seems rather dry and does not help the reader think about the problem.

Author's Response to Decision Letter for (RSOB-20-0061.R0)

See Appendix A.

Decision letter (RSOB-20-0061.R1)

23-Apr-2020

Dear Dr Vincent

We are pleased to inform you that your manuscript entitled "The rise and rise of mitochondrial DNA mutations" has been accepted by the Editor for publication in Open Biology.

Sincerely,
The Open Biology Team
mailto: openbiology@royalsociety.org

Appendix A

Wellcome Centre for Mitochondrial Research
Institute of Neuroscience
Newcastle University
Medical School
Newcastle upon Tyne NE2 4HH
Website: www.newcastle-mitochondria.com/

Director: Professor Sir Doug Turnbull

23rd April 2020

Open Biology

Dear Editor,

We thank you for handling our review “**The rise and rise of mitochondrial DNA mutations**”. We appreciate the reviewers’ helpful comments and have responded to these in a point by point response. We have adjusted the manuscript to take these points into consideration and in particular to include our views on the most likely mechanism of clonal expansion. We hope that you find these changes sufficient but if you would like us to make any further edits, please let us know.

Thank you for accepting our review and we look forward to seeing it published in due course.

Yours sincerely,

Dr. Amy Vincent

Professor Sir Doug Turnbull

Point by point response to Reviewers' comments

Comments 1:

The review provides a good overview of the current understanding of mtDNA accumulation in cells. It does a good job of providing information to the reader, but the writing is somewhat dry. Our understanding of clonal expansion is still incomplete. This review summarizes the current models. It would be helpful if the authors pointed out if they favor a particular model, or otherwise help the reader to think about this problem.

Response 1:

We thank the reviewer for seeing the value in this review. We appreciate the suggestion that we should provide our views on the favoured model of clonal expansion. We believe this to be both tissue and mutation specific and we have explained this and highlighted this also.

Comment 2:

In first paragraph, the circular and multi-copy nature of mtDNA is pointed out as evidence of the endosymbiotic origin of mitochondria. It would be helpful to point out the sequence analysis indicating phylogenetic similarity.

Response 2:

Thank you for this suggestion we have now added a sentence about the sequence analysis as follows:

“Mitochondria are thought to be descended from α -proteobacterium due to sequence similarities between the genome and mitochondrial DNA (mtDNA) (Esser et al. 2004).”

Comment 3:

Page 6; It is pointed out that clonal expansion of mtDNA deletions cannot be explained by random drift. This idea is important and should be explained in more detail. What feature cannot be explained by random drift?

Response 3:

This is an important and we do in fact discuss this in the section on mtDNA deletions later in the review. We could not find this reference on page six, but have inserted an explanation into sentence stating similar on page 4 as follows:

“In comparison to mtDNA point mutations, the clonal expansion of mtDNA deletions does not yet seem to be fully explained by random genetic drift since results from random genetic drift modelling work to date do not accurately predict the levels of mtDNA deletions observed in SN neurons and muscle fibres (discussed further below).”

Comment 4:

Page 6: in discussing the negative feedback model for mtDNA deletions, why can't this model be applied to point mutations that reduce expression of the same proteins?

Response 4:

The reviewer is right that this could be applied to point mutations reducing the expression of the proteins. However, the reason this was not discussed is two-fold, Kowald et al do not suggest this and it has already been established that random genetic drift explains the clonal expansion of mtDNA point mutations it is unlikely that this is contributing significantly to the clonal expansion. However, we have added a sentence to acknowledge this possibility as follows:

“Both the positive transcriptional feedback loop and perinuclear niche hypothesis could feasibly contribute to clonal expansion of mtDNA point mutations in mitotic tissues also, however evidence suggests these selective pressures are not needed for clonal expansion.”

Comment 5:

Page 6: In discussing the nuclear proximity model--how would that mechanism favor mtDNA deletions? Why would genomes with deletions be near the nucleus?

Response 5:

We thank the review for asking this. The nuclear proximity model would simply favour sporadic mutations that have arisen in the perinuclear mitochondria. At least in muscle where the focal deficiency was observed mtDNA deletions are the sporadic mutations that are most frequently reported. Although point mutations are studied less frequently in muscle, so therefore it is possible that this mechanism would also promote clonal expansion of mtDNA point mutations also and this should be investigated. We have now explained the possible preference for mtDNA deletions in the perinuclear region as follows:

“Such stress signalling may include a reduction in ATP/ADP or NAD⁺/NADH ratios, upregulation of the integrated mitochondrial stress response (Kahn et al. 2018), or as suggested in the original investigation upregulation of mitochondrial biogenesis via the unfolded protein response (Vincent et al. 2018). Furthermore, if mtDNA replication is higher in the perinuclear region of muscle fibres as previously reported in HeLa cells (Davis and Clayton, 1996), this may also lead to a higher frequency of replication errors and deletion formation in the perinuclear mitochondria. The perinuclear hypothesis would likely favour sporadic mutations arising in close proximity to the nucleus, and therefore it is possible that it would also favour mtDNA point mutations. However, point mutations are less commonly investigated in muscle and further work would need to be completed to investigate this.”

Comment 6:

Instead of simply stating the current models of clonal expansion, it would be helpful if the authors pointed out their favorite view and why. Otherwise the review seems rather dry and does not help the reader think about the problem.

Response 6:

We thank the reviewer for this suggestion we have now explained our own thinking on this as follows:

“In muscle the presence of perinuclear foci of mitochondrial dysfunction suggests that the nuclei play an important role in the accumulation of mtDNA deletions and mitochondrial dysfunction. Therefore, the perinuclear niche hypothesis shows promise for explaining how these mtDNA deletions accumulate. However, it will be necessary to understand the relative contributions of mtDNA replication and mito-nuclear signalling to this process and to systematically compare this to random genetic drift using in silico modelling (discussed in more detail below). The structure of neurons would favour a situation where the majority of mitochondrial replication would occur in the perinuclear area, and thus would also support a similar perinuclear niche hypothesis for the clonal expansion of mtDNA deletions to that proposed for muscle. However, this is less definitively described in neurons than muscle and also needs to take into consideration the vastly different movement and dynamics of mitochondria within these two cell types. This may explain why perinuclear focal deficiency has not to date been reported in neurons, however it is still possible that such focal deficiency occurs and is simply less frequent or more challenging to find.”